# Do What Nature Did To Us: Evolving Plastic Recurrent Neural Networks For Task Generalization

## Abstract

While artificial neural networks (ANNs) have been widely adopted in machine learning, the gaps between ANNs and biological neural networks (BNNs) are receiving increasing concern. In this paper, we propose a framework named as *Evolutionary Plastic Recurrent Neural Networks* (EPRNN). Inspired by BNN, EPRNN composes Evolution Strategies, Plasticity Rules, and Recursion-based Learning in one meta-learning framework for generalization to different tasks. More specifically, EPRNN incorporates nested loops for meta-learning — an outer loop searches for optimal initial parameters of the neural network and learning rules; an inner loop adapts to specific tasks. In the inner loop of EPRNN, we effectively attain both long-term and short-term memory by forging plasticity with recursion-based learning mechanisms, both of which are believed to be responsible for memristance in BNNs. The inner-loop setting closely simulates BNNs, which neither use gradient-based optimization nor require the exact forms of learning objectives. To evaluate the performance of EPRNN, we carry out extensive experiments in two groups of tasks: *Sequence Predicting*, and *Wheeled Robot Navigating*. The experiment results demonstrate the unique advantage of EPRNN compared to state-of-the-arts based on plasticity and recursion while yielding comparably good performance against deep learning-based approaches in the tasks. The experiment results suggest the potential of EPRNN to generalize to a variety of tasks and encourage more efforts in plasticity and recursion-based learning mechanisms.

## 1 Introduction

ANNs have achieved great success in handling machine learning tasks. Despite being initially inspired by Biological Neural Networks (BNNs), there are apparent gaps between ANNs and BNNs. Mainstream ANNs use gradient-based optimizers to minimize learning objectives. Shreds of evidence show that BNNs learn through plasticity (Gerstner et al., 1993) without explicit learning objectives, among which Hebb's rule (Hebb, 1949) is most well known. Though gradient descent methods are the most efficient optimizers for ANNs, their side effects are also noticed, including the problems of catastrophic forgetting, over-consumption of data, and the requirement for manual efforts in designing objective functions. Those challenges are becoming an essential impedance to the further development of machine intelligence.

Recent studies show the learning mechanisms of BNNs, such as plasticity (Soltoggio et al., 2008; Najarro & Risi, 2020) and model-based learning (Santoro et al., 2016; Mishra et al., 2018), under appropriate meta-parameter optimization, can be effective alternative for task generalization in ANNs. Unlike gradient-based methods, these mechanisms simulate the learning behaviors of BNNs and don't require any explicit-form learning objectives. More recently, authors in (Miconi et al., 2019) proposed a plastic recurrent neural network for lifelong learning of ANNs, where implements Hebbian plasticity with differentiable objectives and gradient-based optimization. Though the above studies have investigated learning of ANNs using the two mechanisms derived from BNNs with gradient-based methods (Miconi et al., 2019) optionally, in this work, we aim at further verify the path of discovering those rules evolutionarily, simulating that of BNNs.

**Backgrounds.** Though learning in BNNs has not been fully understood, some of the learning mechanisms and rules, such as plasticity (Gerstner et al., 1993) and recursion (Pollen, 2003), have been observed in brains and adopted by ANNs. Typically, Model-based learning employs recurrent neural networks (*RNN*), *LSTM* (Hochreiter & Schmidhuber, 1997), and self-attention (Mishra et al., 2018; Chen et al., 2021) layers as learners. Learning is based on memories within the feed-forward pass. The information is updated in the hidden states instead of the parameters. Model-based learners are found to be sample efficient in generalized supervised tasks (Santoro et al., 2016), zero-shot generalization in language (Brown et al., 2020), and reinforcement learning (Mishra et al., 2018; Chen et al., 2021) when compared with various type of gradient descent methods. So far, among model-based learners, though self-attention-based learners such as *Transformers* have state-of-the-art performance, the $O(T^2)$ (where $T$ is the sequence length) makes them only available to relatively short sequences. On the other hand, recurrent learners such as *RNN* and *LSTM* have $O(T)$ inference costs but suffer from poor asymptotic performances. That is, when sequences are getting longer, performances no longer improve or even deteriorate. It is partly due to the limitation of the memory spaces. For instance, an recurrent neural network of hidden size $n$ has a memory of $O(n)$. In contrast, its parameters scale with $O(n^2)$, making parameter-updating more powerful as learning mechanisms than recursion-only.

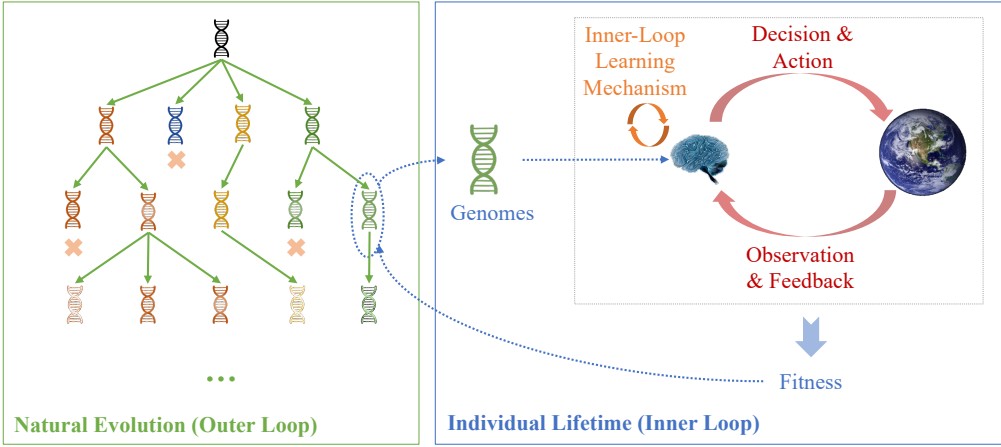

Figure 1: An illustration of the natural evolution: The evolution takes place in the outer loop, where the genomes are mutated and selected, and the population either thrive or become extinct based on the *Fitness*. Lifetime of each individual composes the inner loop. At the beginning, the genomes decide the learning mechanisms and initial neural configurations of the brain in the new born life. As the neural networks interact with the environment through actions and observations, its connections and hidden neuron states are further updated to better adapt to the environment. Plasticity are believed to be important part of the learning mechanisms. The fitness depends on the learning and adapting capability of each individual.

In addition to recursion-based learning, evolving plasticity (Soltoggio et al., 2008; 2018; Lindsey & Litwin-Kumar, 2020; Yaman et al., 2021) has been proposed to reproduce the natural evolution and plasticity in simulation, as shown in Figure 1. Implementing plasticity is not straightforward; unlike gradient-based learning methods, plastic rules are not universal but have to be optimized beforehand, which is not possible without a further outer-loop optimizer over the inner-loop learning. Evolutionary algorithms (Zhang et al., 2011; Salimans et al., 2017a) are typically applied in the outer loop to search for meta-parameters shaping the learning rules, which can be regarded as information carried by genomes during evolution. Those optimized plasticity rules are then applied in the inner loop to further tune NN's parameter for better adaptions to the environment. Another line of works tries to bring gradient-based learning algorithms to plasticity rule optimization (Miconi et al., 2018; 2019). It is found that evolution can be more efficient in cases of long-horizon in reinforcement learning (Salimans et al., 2017b; Stanley, 2019).

**Our Works.** Inspired by the previous works (Cabessa & Siegelmann, 2014; Miconi et al., 2018; 2019) that improve recursive neural networks using plastic rules for capacity of learning, we propose

| Methods | Inner Loop | Outer Loop |
|---|---|---|
| **Memory Augmented NN**, (Santoro et al., 2016) | Recursion | Gradient |
| **MAML**, (Finn et al., 2017) | Gradient | Gradient |
| **Conditional Neural Processes**, (Garnelo et al., 2018) | Average | Gradient |
| **SNAIL**, (Mishra et al., 2018) | Attention | Gradient |
| **ES-MAML**, (Song et al., 2019) | Gradient, Hill Climbing | Evolution |
| **EPMLP**, (Najarro & Risi, 2020) | Plasticity | Evolution |
| **Differential Platicity**, (Miconi et al., 2018) | Plasticity & Recursion | Gradient |
| **Backpropamine**, (Miconi et al., 2019) | Plasticity & Recursion | Gradient |
| **EPRNN** (ours) | Plasticity & Recursion | Evolution |

Table 1: A brief review of meta-learning methods

a novel meta-learning framework namely _Evolutionary Plastic Recurrent Neural Networks_ (EPRNN) for task generalization. Specifically, this work makes contributions as follows.

- We study the potential of learning plasticity and recursion rules through the natural evolution in task generation. We show that recursion and plasticity-based rules can surpass gradient-based methods as inner-loop learners.
- We present investigations and analyses on the learned rules and parameters, showing that the learning framework discovers plasticity rules that effectively update the connection weights according to the learning tasks. The differences between the transformation of hidden states and parameters are also shown, verifying the efficacy of combining recursion with plasticity.

The most relevant works to our study are (Miconi et al., 2018; 2019; Lindsey & Litwin-Kumar, 2020; Yaman et al., 2021). Compared to (Miconi et al., 2018) that leverage gradient oracles to efficiently search plastic rules, the proposed EPRNN could work well in gradient-free settings. Compared to (Lindsey & Litwin-Kumar, 2020; Yaman et al., 2021) that use evolutionary strategies to learn plastic rules, EPRNN also incorporates an RNN-based inner loop for recursion-based learning. The work (Miconi et al., 2019) also uses recursion (i.e., RNN) and differentiable plasticity in nested loops to train self-modifying neural networks, EPRNN replaces the outer loop with evolutionary strategies to generalize tasks with non-differentiable objectives. Though EPRNN is not as competitive as gradient-based methods, which can optimize advanced neural networks with large datasets, our work still demonstrates the potential of using plasticity and recursion for meta-learning through natural evolution.

## 2    RELATED WORKS

**Meta-Learning.** Meta-learning aims at building learning machines that gain experience using task-specific data over the distribution of tasks. Inspired by human and animal brains that are born with both embedded skills and the capability of acquiring new skills, meta-learning implements two nested learning loops: The outer learning loops optimize the meta-parameters that typically involves initial parameters (Finn et al., 2017; Song et al., 2019), learning rules (Zoph & Le, 2017; Najarro & Risi, 2020; Pedersen & Risi, 2021), model structures (Soltoggio et al., 2008; Li & Malik, 2016) and all of three (Real et al., 2020) over distribution of tasks; The inner learning loops adapt the model to specific tasks by utilizing those meta-parameters. According to different inner-loop optimizers, we roughly classify the methods into _model-based_ and _parameter-updating_ methods. The _model-based_ methods do not update the parameters in the inner-loop, where only hidden states is updated (e.g., recursion); The _parameter-updating_ methods re-modify the connection weights in the inner-loop (e.g., gradient descent (MAML), plasticity). From this point of view, our method can be classified into both groups. A brief review of the typical meta-learning paradigms is presented in Table 1.

**Plasticity-based Learning.** The proposal of the learning mechanism of BNNs is initially raised by Hebb's rule (Hebb, 1949), the most prominent part of which is "neurons fire together wire together". It is further polished by Spike Time-Dependent Plasticity (STDP) (Gerstner et al., 1993) indicating that the signal of learning is dependent on the temporal patterns of the presynaptic spike and postsynaptic spike. Learning could also appear in inhibitory connections, also known as anti-Hebbian

(Barlow, 1989). Also, relationships between STDP and memory are investigated (Linares-Barranco & Serrano-Gotarredona, 2009). Since many of those rules are related to spiking neural networks (Ghosh-Dastidar & Adeli, 2009), to apply them to ANNs, simplified rules are proposed (Soltoggio et al., 2008) instead: given the pre-synaptic neuron state $X$ and post-synaptic neuron state $Y$, the connections between $X$ and $Y$ are updated by

$$\delta W = m[A \cdot XY + B \cdot X + C \cdot Y + D], \tag{1}$$

$m$ is the output from neuron modulators that adjust the learning rates of plasticity. Most of the existing rules are sub-classes of Equation 1. For instance, some works neglect the neural modulator $m$ (Najarro & Risi, 2020; Miconi et al., 2018), others have set $B$, $C$, and $D$ to 0 (Miconi et al., 2018; 2019). The learned rule $A, B, C, D$ will inevitably be dependent on initial parameter of $W$, however, learning plastic rules that is not dependent on the initial parameters was also investigated (Najarro & Risi, 2020; Yaman et al., 2021).

## 3 ALGORITHMS

**Problem Settings**. We consider an agent (learner) that is dependent on meta-parameter $\theta$. It has the capability of adapting itself to a distribution of tasks $T_j \in \mathcal{T}$ by interacting with the environment $T_j$ through observation $i_t$ and action $a_t$. In K-shot learning, the agent is allowed to first observe samples of length $K$ (this stage can be referred as *meta-training-training*, see Beaulieu et al. (2020)), then its fitness is calculated in *meta-training-testing* rollouts. In Generalized Supervised Learning tasks (**GSL**), the observations typically include features ($x_t$) and labels ($y_t$) in *meta-training-training* stage ($i_t = \{x_t, y_t\}$), and the labels are left out for predicting in the *meta-training-testing* stage (Santoro et al., 2016; Garnelo et al., 2018). In Generalized Reinforcement Learning tasks (**GRL**), the observations typically include states ($s_t$), actions ($a_{t-1}$), and feedbacks ($r_{t-1}$) ($i_t = \{s_t, a_{t-1}, r_{t-1}\}$, sometimes $r_{t-1}$ can not be observed) (Mishra et al., 2018). The goal of *meta-training* is to optimize $\theta$ such that the agent achieves higher fitness in *meta-training-testing*. In *meta-testing*, similarly, the learned parameters are given *meta-testing-training* and *meta-testing-testing* in order, the performances in *meta-testing-testing* are evaluated.

**Plastic Recurrent Neural Networks (PRNN)**. Given a sequence of observations $i_1, ..., i_t, ...$, we first consider an recurrent neural network (*RNN*) that propagates forward and yields sequence of outputs $a_t$ following:

$$h_{t+1} = \sigma(W_t \cdot h_t + W_i \cdot i_t + b), \tag{2}$$

$$a_t = f(h_{t+1}) \tag{3}$$

where $h_t$ is the hidden states at step $t$. In PRNN, we kept $W_i$ stationary, but we set $W_t$ to be plastic, so that we add a subscript $t$ to mark different $W_t$ at different steps. Regarding $h_t$ as pre-synaptic neuron states, and $h_{t+1}$ as post-synaptic neuron states, by applying Equation 1, we update $W_t$ with:

$$W_{t+1} = W_t + \delta W_t \tag{4}$$

$$\delta W_t = W_A \odot (\hat{h}_{t+1} \cdot h_t^{\mathrm{T}}) + W_B \odot (m_t \cdot h_t^{\mathrm{T}}) + W_C \odot (\hat{h}_{t+1} \cdot \mathbf{1}^{\mathrm{T}}) + W_D \cdot m_t \tag{5}$$

$$\hat{h}_{t+1} = m_t \odot h_{t+1}, \tag{6}$$

where we use $\odot$ and $\cdot$ to represent "element-wise multiplication" and "matrix multiplication" respectively. $h$ and $\mathbf{1}$ are column vectors. $W_A, W_B, W_C, W_D$ are collection of plastic rules of $A, B, C, D$ from Equation 1, which has the same shape as $W_t$. $m_t$ is the neural modulators that adjusts the learning rates of plasticity. We calculate $m_t$ by applying a neuron modulating layer denoted with:

$$m_t = \sigma(W_h^{(m)} \cdot h_t + W_i^{(m)} \cdot i_t + b^{(m)}). \tag{7}$$

A sketch of PRNN is presented in Figure 2. The main difference between PRNN and naive *RNN* is that PRNN updates both the hidden states and the connection weights during the forward pass.

**Evolving PRNN**. Given task $T_j \in \mathcal{T}$, by continuously applying Equation 2 to 7 over *meta-training-training* and *meta-training-testing*, the fitness is eventually dependent on the initial parameters, learning rules, and the sampled task $T$, which is denoted as:

$$Fit(\theta, T) = Fitness(i_{K+1}, a_{K+1}, i_{K+2}, a_{K+2}, ...,) \tag{8}$$

$$W_i, W_0, W_A, W_B, W_C, W_D, W_h^{(m)}, W_i^{(m)}, b, b^{(m)} \in \theta \tag{9}$$

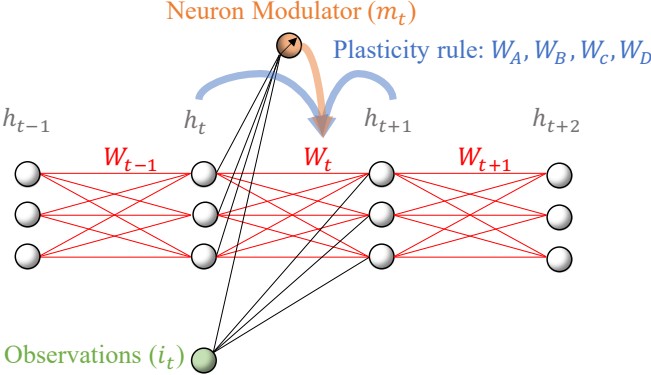

Figure 2: A sketch of the information flow in plastic recurrent neural networks (PRNN). The red connections are plastic, the black connections are static.

Following Evolution Strategies (ES) (Salimans et al., 2017a), in $k$th outer-loop iteration, we sample different tasks from $\mathcal{T}$, and meta-parameters $\theta_{k,i}$ ($i \in [1, n]$) from the neighbourhoods of $\theta_k$. We evaluate the fitness of sampled meta-parameters, and update the meta-parameters by applying:

$$\theta_{k+1} = \theta_k + \alpha \frac{1}{n} \sum_{i=1}^{n} Fit(\theta_{k,i}, T_k)(\theta_{k,i} - \theta_k) \tag{10}$$

**Why Recurrent Neural Networks?** As stated in Equation 1, plasticity in feed-forward-only NNs allows NNs to gain experiences from single-frame observation only. In cases of non-sequential GSL, the plasticity has chances to tune the connection weights to the specific task by relying on observing one single frame of data ($i_t = \{x_t, y_t\}$), since its information of the feature and the supervision is complete. However, in general cases, learning can be effective without summarizing sequences of observations. For instances, a human driver getting used to a new car through continuously interacting and observing. Moreover, in GRL, there are time lag between the observation of states and feed-backs. Recursion helps to summarize historical observations to give the correct update for the connection weights.

Although, compared with naive *RNN*, there are obviously bunches of more sophisticated neural structure such as *GRU* and *LSTM*, we believe it is more desirable to start from simplest recurrent structure to study the potential of combining recursion and plasticity.

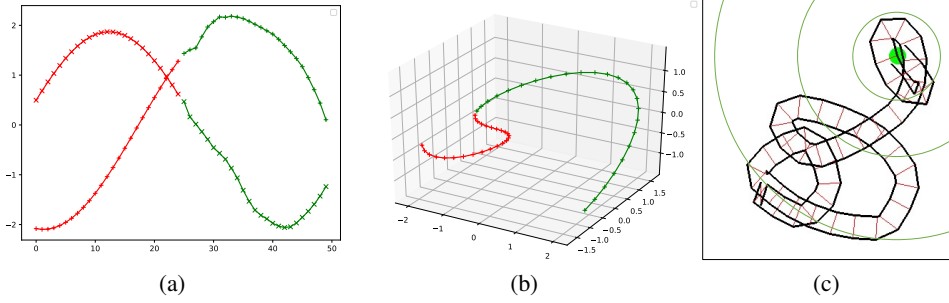

Figure 3: Demonstration of the tasks.(a) Two tasks sampled from *Sequence Predicting* (l=1,K=25,N=25), the red lines are training sets and the green lines are testing sets. (b) One task sampled from *Sequence Predicting* (l=3,K=25,N=25). (c) A trajectory generated by an agent in a *Wheeled Robot Navigating* task.

## 4 EXPERIMENTS

### 4.1 TASKS FOR EVALUATION

In generalized tasks, we have each task $T_j$ dependent on some configuration parameters that are hidden from the agent. Below we introduce two groups of generalized tasks that we experiment on.

*Sequence Predicting* (Generalized Supervised Learning tasks). We randomly generate sequences of vectors $\boldsymbol{y}(t) = (y_1(t), ..., y_l(t))$, where $y_i(t) = A_i sin(\frac{2\pi t}{n_i} + \phi_i)$, $t = 1, 2, 3, ...$ and $A_i \sim \mathcal{U}(1, 3)$, $n_i \sim \mathcal{U}(10, 100)$, $\phi_i \sim \mathcal{U}(0, 2\pi)$, and $\mathcal{U}(a, b)$ represents the uniform distribution between $a$ and $b$. $A$, $n$ and $\phi$ are hidden from the agent. The front part of the sequence $\boldsymbol{y}(1), \boldsymbol{y}(2), ..., \boldsymbol{y}(K)$ is exposed to the agent, and the left part $\boldsymbol{y}(K + 1), \boldsymbol{y}(K + 2), ..., \boldsymbol{y}(N)$ is to be predicted. The fitness is the opposite of mean square error (MSE) between the predicted sequence and the ground truth. We test the methods for comparison in four groups of tasks including $(l = 1, K = 10, N = 20)$, $(l = 1, K = 25, N = 50)$, $(l = 3, K = 10, N = 20)$, and $(l = 3, K = 25, N = 50)$ (see Figure 3(a)(b)).

*Wheeled Robot Navigating* (Generalized Reinforcement Learning tasks). The agent is to navigate a two-wheeled robot to a randomly generated goal in 2-D space $\boldsymbol{g} = (g_x, g_y)$. We assume that there is a signal transmitter on the goal and a receiver on the robot. The robot observes the signal intensity decided by $A_t = A_0 - k \cdot log(d_t/d_0) + \epsilon$ (inspired by the attenuation for electromagnetic wave, see Friis (1946)), where $d_t$ is the current distance between the robot and the goal, $\epsilon \sim \mathcal{N}(0, \sigma)$ is the white noise in the observation. $g_x, g_y, A_0, k$ are environment related configurations that is hidden from the agent. For each task, we sample configurations by $g_x, g_y \sim \mathcal{U}(-0.5, 0.5)$, $A_0 \sim \mathcal{U}(0.5, 2.0)$, and $k \sim \mathcal{U}(0.1, 0.5)$. The action is the rotation speed of its two wheels that controls the orientation and velocity of the robot. The reward at each step is $r_t = -d_t$, an episode terminates when the robot approaches the goal or steps reaches the maximum of 100. We also hide its own position and orientation from the agent, such that the agent relies on recording its own action the signal strength $A_t$ to locate itself. We investigate three types of navigating circumstances with different level of noises in the observed signal: Low Noise ($\sigma = 0.01$), Median Noise ($\sigma = 0.05$), and High Noise ($\sigma = 0.2$) (see Figure 3(c)).

### 4.2 EXPERIMENT SETTINGS

We add the following methods into comparison, the methods share exactly the same outer loop and differ in the inner loop.

- **ES-MAML** (Song et al., 2019) : We use four gradient descent steps in the inner loop, the learning rate of each step is treated as meta-parameters which is to be optimized by the outer loop. As MAML can not utilize instant observation in zero-shot case, we show results of both **ES-MAML (zero-shot)** and **ES-MAML (one-rollout)** in *Wheeled Robot Navigating*. Except for **ES-MAML**, the other methods are measured with zero-shot *meta-testing* score only in *Wheeled Robot Navigating*.

- **ES-RNN**: Vanilla *RNN* as the inner loop learner.

- **ES-LSTM**: Long Short Term Memory (Hochreiter & Schmidhuber, 1997) as the inner loop learner.

- **EPMLP** (Soltoggio et al., 2018): Multi-Layer Perceptrons (MLP) with plasticity rules implemented.

- **EPMLP (Random)** (Najarro & Risi, 2020): The main difference with **EPMLP** is randomly setting the parameters of the plastic layers at the beginning of each inner loop instead of using fixed trainable initial parameters.

- **EPRNN (w/o m)**: Removing neuron modulator ($m_t$) from the plasticity rule in PRNN.

We add additional non-plastic fully connected layers before and after the plastic layers to increase the representation capability of the models. For fairness, we kept those layers identical for all compared methods. For every 100 outer loops in *meta training*, we add an extra *meta testing* epoch, evaluating the average fitness of current meta-parameters on testing tasks. Each run includes 15000 outer

loops and 150 meta-testing epochs. Each result is concluded from independent 3 runs. Our code[1] relies on PARL[2]for parallelization. We leave the detailed illustration of the model structures and hyper-parameters in the Appendices.

## 4.3 RESULTS

We present the experiment results in Figure 4 (In Table 2, and Table 3, we also list the summarized performance by averaging the Top-3 *meta-testing* scores in the latest 10 *meta-testing* epochs of each run over 3 independent runs). Generally, we can conclude that PRNN performs substantially better when compared with naive *RNN*. In some cases, it even produces better results compared with *LSTM*, despite the simpler model architecture. It is also interesting to notice that the gap between *RNN* and PRNN are smaller in shorter sequences or low-noise environments, but larger in more challenging tasks with longer sequence or higher noise (In *Wheeled Robot Navigating* tasks, higher noise pushes the agent to maintain a longer memory in order to filter the noise and figure out the way to goal). This phenomenon reaffirms the lack of long-term memories in *RNN*, and shows that PRNN significantly improves this drawback. Comparing **EPRNN (w/o m)** with *RNN* and **EPRNN** clearly demonstrates that simple ABCD rule (without the neural modulator) may also work to some extent, but introducing the neuron modulator $m_t$ can further benefit the learner.

Table 2: Summarized performances comparison in *Sequence Predicting* tasks

| Methods | l=1,K=10,N=20 | l=1,K=25,N=50 | l=3,K=10,N=20 | l=3,K=25,N=50 |
|---|---|---|---|---|
| **ES-RNN** | $-0.385 \pm 0.060$ | $-1.228 \pm 0.191$ | $-1.273 \pm 0.009$ | $-1.811 \pm 0.015$ |
| **ES-LSTM** | $-0.165 \pm 0.014$ | $-0.283 \pm 0.013$ | $-1.229 \pm 0.010$ | $-1.475 \pm 0.016$ |
| **ES-MAML** | $-1.452 \pm 0.292$ | $-1.747 \pm 0.064$ | $-1.218 \pm 0.013$ | $-1.796 \pm 0.013$ |
| **EPMLP** | $-0.732 \pm 0.031$ | $-1.185 \pm 0.045$ | $-1.339 \pm 0.004$ | $-1.735 \pm 0.013$ |
| **EPMLP-(Random)** | $-1.233 \pm 0.115$ | $-1.319 \pm 0.035$ | $-1.489 \pm 0.023$ | $-1.788 \pm 0.003$ |
| **EPRNN** | $\mathbf{-0.114 \pm 0.012}$ | $\mathbf{-0.208 \pm 0.025}$ | $\mathbf{-1.107 \pm 0.015}$ | $\mathbf{-1.430 \pm 0.019}$ |
| **EPRNN (w/o m)** | $-0.135 \pm 0.026$ | $-0.351 \pm 0.058$ | $-1.128 \pm 0.014$ | $-1.520 \pm 0.041$ |

Table 3: Summarized performances comparison in *Wheeled Robot Navigating* tasks

| **Methods** | Low Noise ($\sigma = 0.01$) | Median Noise ($\sigma = 0.05$) | High Noise ($\sigma = 0.2$) |
|---|---|---|---|
| **ES-RNN** | $-16.90 \pm 1.30$ | $-18.99 \pm 0.14$ | $-31.23 \pm 3.95$ |
| **ES-LSTM** | $-14.04 \pm 0.08$ | $-15.50 \pm 0.58$ | $\mathbf{-22.53 \pm 0.45}$ |
| **ES-MAML (zero-shot)** | $-37.18 \pm 0.45$ | $-37.91 \pm 0.80$ | $-37.90 \pm 0.05$ |
| **ES-MAML (1 rollout)** | $-23.10 \pm 0.46$ | $-29.21 \pm 0.46$ | $-37.12 \pm 0.12$ |
| **EPMLP** | $-21.24 \pm 6.08$ | $-16.65 \pm 0.39$ | $-23.75 \pm 2.02$ |
| **EPMLP (Random)** | $-13.02 \pm 0.19$ | $-18.25 \pm 0.23$ | $-28.85 \pm 1.1$ |
| **EPRNN** | $\mathbf{-12.71 \pm 0.75}$ | $\mathbf{-15.07 \pm 0.03}$ | $-23.79 \pm 0.55$ |
| **EPRNN (w/o m)** | $-14.93 \pm 0.49$ | $-16.72 \pm 0.33$ | $-24.11 \pm 0.34$ |

Among plasticity based methods, we show that recursion is more advantageous than MLP in evaluated tasks. It is also worth noticing that **EPMLP** and **EPMLP (Random)** perform steadily beyond the gradient-based learner (**ES-MAML**). In **ES-MAML**, the gradient can only be calculated after an episode is completed, while **EPMLP** is able to perform sequential learning even though no feedback is available during its life time. This demonstrates the possibility of surpassing human-designed gradient-based learning rules with automatically learned unsupervised rules. Moreover, comparison between **EPMLP** and **EPMLP (Random)** validate the proposal of Najarro & Risi (2020), implying the possibility of discovering global plastic learning rules instead of rules coupled with the initial

---

[1]https://github.com/WorldEditors/EvolvingPRNN
[2]https://github.com/PaddlePaddle/PARL

parameters. Yet we see optimization of the initial parameters is still advantageous, which can also be validated by evidences in nature that the newborn lives already have certain embedded skills (e.g., Newborn human babies have reflexes of suckling and grasping; Foals can stand shortly after being born).

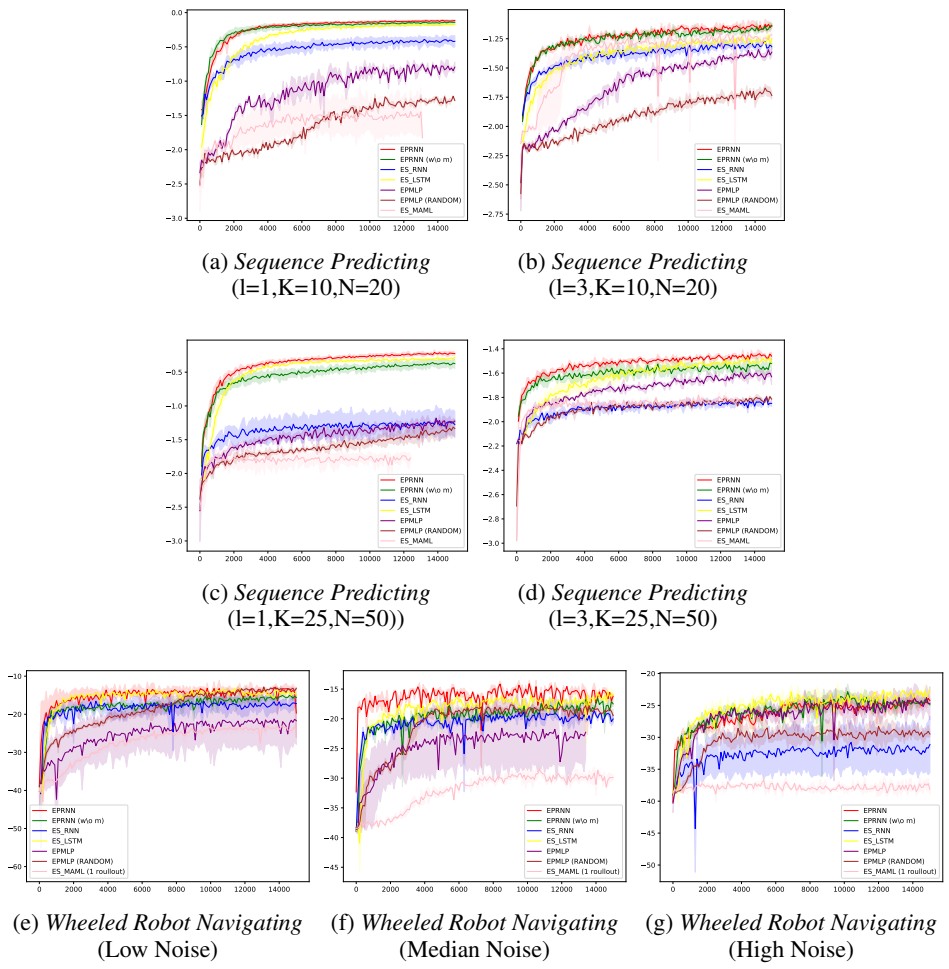

(a) *Sequence Predicting*
(l=1,K=10,N=20)

(b) *Sequence Predicting*
(l=3,K=10,N=20)

(c) *Sequence Predicting*
(l=1,K=25,N=50))

(d) *Sequence Predicting*
(l=3,K=25,N=50)

(e) *Wheeled Robot Navigating*
(Low Noise)

(f) *Wheeled Robot Navigating*
(Median Noise)

(g) *Wheeled Robot Navigating*
(High Noise)

Figure 4: Plotting *meta-testing* scores against *meta-training* iterations.

## 4.4 ANALYSIS AND DISCUSSION

To investigate whether plasticity rules update the connection weights as expected, we test the trained model with different tasks and record the updating trajectories of plastic connection weights $W_t$ and hidden states $h_t$. We run t-SNE visualization to map those tensors ($W_t$s and $h_t$s) to 2-D space and show their trajectories in Figure 5. The connection weights $W_t$ typically start from the same position and gradually move in different directions depending on the task configurations. The final weights effectively capture environmental configuration that was hidden from the agent. Particularly, in Figure 5(d) for *Wheeled Robot Navigating* tasks, we can see that $W_t$ captures only the signal transmission patterns ($A_0, k$), but neglects the position of the goal ($g_x, g_y$). We guess that $A_0, k$ are important stationary patterns that helps the agent to interpret the observed signal strength ($A_t$), while the absolute position of goal is less important as its relative position to the robot is changing continuously. This demonstrates that plasticity has performed meaningful updates on the weights of the connections depending on the tasks. On the other hand, the hidden states (Figure 5(c)(d)(e)(h)) are noisier and less distinguishable among different tasks. The small repeated circles in the trajectory of hidden states might correspond to short-term patterns such as the periodicity of *Sequence Predicting* task and the circular movements in *Wheeled Robot Navigation* tasks. (see Appendices A.2)

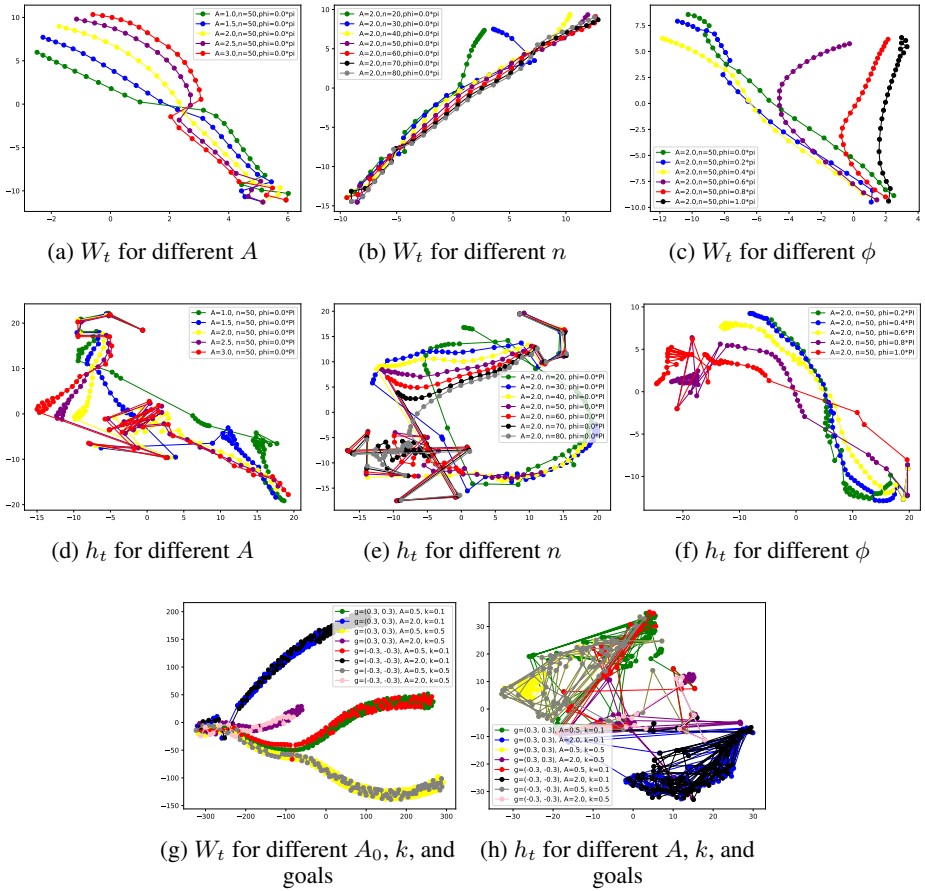

(a) $W_t$ for different $A$     (b) $W_t$ for different $n$     (c) $W_t$ for different $\phi$

(d) $h_t$ for different $A$     (e) $h_t$ for different $n$     (f) $h_t$ for different $\phi$

(g) $W_t$ for different $A_0$, $k$, and goals     (h) $h_t$ for different $A$, $k$, and goals

Figure 5: t-SNE visualization of the transformation of the connection weights ($W_t$) and hidden states ($h_t$) in the inner loop. (a)(b)(c)(d)(e)(f) are from *Sequence Predicting* (l=1,K=25,N=50) task, (g)(h) are from *Wheeled Robot Navigating* (Low Noise) task.

Note that we did not compare EPRNN with many gradient-based solutions such as differentiable plasticity (Miconi et al., 2018; 2019), since the classic Hebbian plasticity has been observed and confirmed from neurons for more than half century (Hebb, 1949) — there is not evidences that the nature do differentiation or backpropagation (Lillicrap et al., 2020) to update plastic rules. The purpose of our work is to reveal the potential of natural evolution of plasticity, and we don't hope to claim that EPRNN could outperform gradient-based solutions on machine learning tasks.

## 5 CONCLUSIONS

In this paper we present EPRNN a nature-inspired learning framework composed of Evolution Strategies, Plastic rules, and Recursion. Experiment results show that plasticity can be effectively forged with recursion to enhance the learning capability. The proposed framework can achieve equivalent or even better performances compared with more sophisticated neural structures, by applying the simplest recurrent neural structures. Moreover, we also show that under proper meta parameters, plasticity has a chance to surpass gradient descent methods.

We believe the learning framework of Figure 1 can be extended to more sophisticated plastic rules and model structures, uncovering better learners in the future. Also, it would be more interesting if such framework can be validated in more complex environments such as natural language processing tasks and vision-related tasks. Finally, we are looking forward that this work can shed light to new paradigm of building intelligent machines and inspire more efforts in this line of research.

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

# A   APPENDICES

## A.1   MODEL ARCHITECTURE AND TRAINING DETAILS

To maintain fairness for comparison, the model architectures of the compared methods are similar, where the differences lie in only one of their layers (Figure 6). We use 3 hidden layers for *Sequence Predicting* tasks and 4 hidden layers for *Wheeled Robot Navigating* tasks. For **ES-MAML**, we replace the PRNN layer with fully connected layer; For **ES-RNN** and **ES-LSTM**, we replace it with *RNN* and *LSTM* respectively; For **EPMLP**, we replace it with plastic fully connected layer, where the plasticity rule is stated by Equation 1, and the neural modulator $m$ is calculated by an additional dense layer with sigmoid activation. The hidden sizes of all the hidden layers are $64$ (for LSTM it is $64$ hidden states and $64$ cell states).

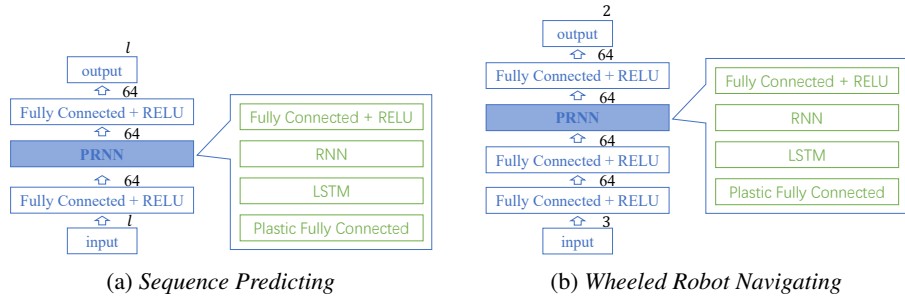

(a) *Sequence Predicting*          (b) *Wheeled Robot Navigating*

Figure 6: A sketch of the model architectures for evaluated tasks

For *Sequence Predicting* tasks, the input observation and the output action has the dimension of $l$. In the *meta-training-training* and *meta-testing-train* stages, we use the ground truth $\boldsymbol{y}_{t-1}$ as input observation; In the the *meta-training-test* and *meta-testing-test* stages, its previous action ($a_{t-1}$) are taken as inputs. For *Wheeled Robot Navigating* tasks, the output action is the control command of its two wheels (a length 2 vector), the input observation is the concatenation of its previous action and the current observed signal intensity ($A_t$).

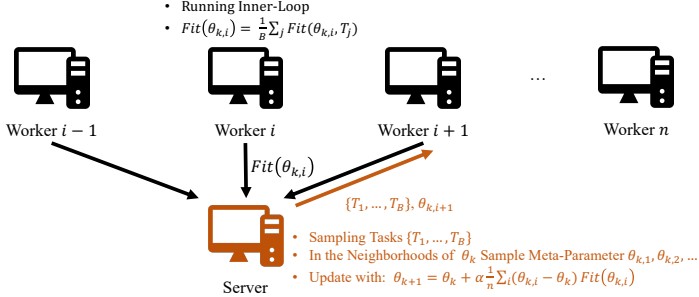

Figure 7: A sketch of the parallel training framework.

The training process is accelerated by utilizing the paralleling mechanism of PARL. We employ 400 workers (400 Intel(R) Xeon(R) CPU E5-2650) running inner-loops for 400 off-springs in each generation of the evolution, and additional 1 CPU to perform evolutionary update (shown in Figure 7(b)). It takes 3 to 12 hours for each run depending on the length of the inner loop and model architectures. Following the previous work (Salimans et al., 2017a), we rank normalize the fitness among 400 workers. The learning rate $\alpha$ is set to $0.2$. The mutation is performed by adding independent Gaussian noises to each parameter. During the *meta-training*, we sample $B$ different tasks and 400 meta-parameters. Each worker run $B$ *meta-training-training* and $B$ *meta-training-testing* given the assigned meta-parameter, then the fitness is averaged. We set $B = 16$ for *Sequence*

*Predicting* tasks, and $B = 4$ for *Wheeled Robot Navigating* tasks. For each *meta-testing* epoch, we evaluated the current meta parameters in newly sampled 1600 tasks.

## A.2 SUPPLEMENTARY EXPERIMENTAL RESULTS AND ANALYSES

We show more t-SNE visualization of the connection weights and hidden states in Figure 8 which can be compared with Figure 5. The transformation of hidden states in **ES-RNN** (Figure 8(a)(b)(c) and (e)) are even less distinguishable among different tasks compared with that of **EPRNN** (Figure 5(d)(e)(f) and (h)), showing naive RNNs fail to capture task configurations. The weights of **EPMLP** can also capture some hidden configurations (Figure 8(d)) but are less accurate compared with that of **EPRNN** (Figure 5(g)). For instance, the gold and grey line share the same hidden configuration of $A = 0.5$ and $k = 0.5$. The connection weights effectively captures this information and the two lines overlap with each other in **EPRNN**, but it does not happen in **EPMLP**. Similar phenomenon can be observed for the red and the green lines ($A = 0.5$ and $k = 0.1$).

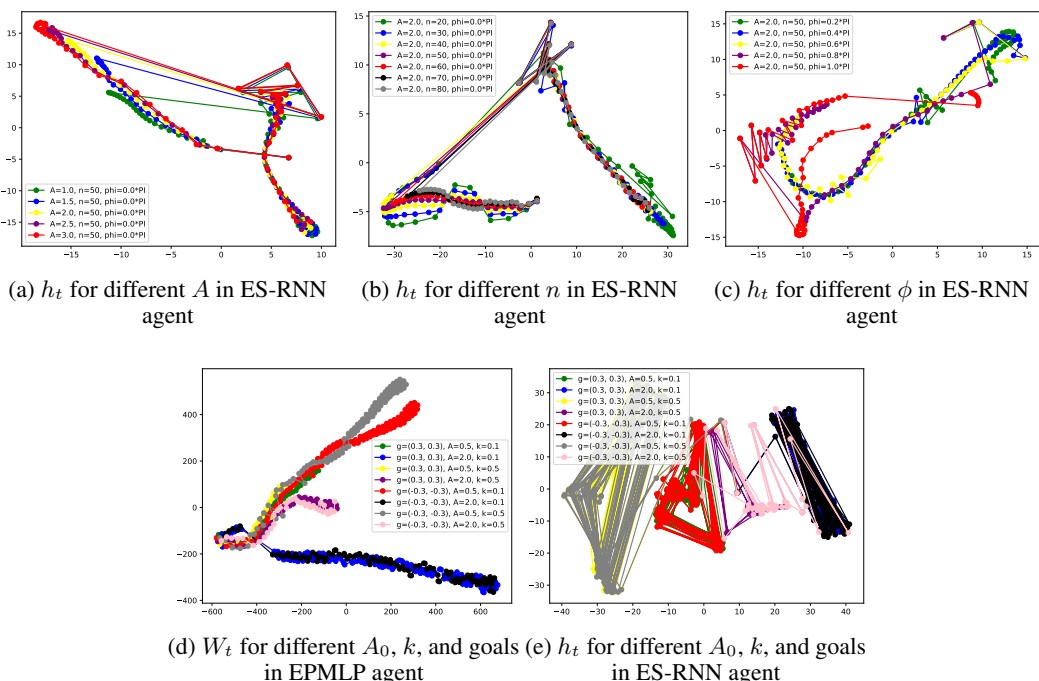

(a) $h_t$ for different $A$ in ES-RNN agent

(b) $h_t$ for different $n$ in ES-RNN agent

(c) $h_t$ for different $\phi$ in ES-RNN agent

(d) $W_t$ for different $A_0$, $k$, and goals in EPMLP agent

(e) $h_t$ for different $A_0$, $k$, and goals in ES-RNN agent

Figure 8: Supplementary t-SNE visualization of the transformation of the connection weights ($W_t$) and hidden states ($h_t$) in the inner loop. (a)(b)(c) are from *Sequence Predicting* (l=1,K=25,N=50) task, (d)(e) are from *Wheeled Robot Navigating* (Low Noise) task.

We also plot the trajectories of the robot in *Wheeled Robot Navigation* tasks. All the agents struggle to find the goal by continuously shifting their directions, which is more frequent in cases of higher observation noise. A reasonable guess is that in higher noise cases, the agents are less confident in their estimations. They have to more frequently adjust their directions to acquire diverse observations for more effective denoising.

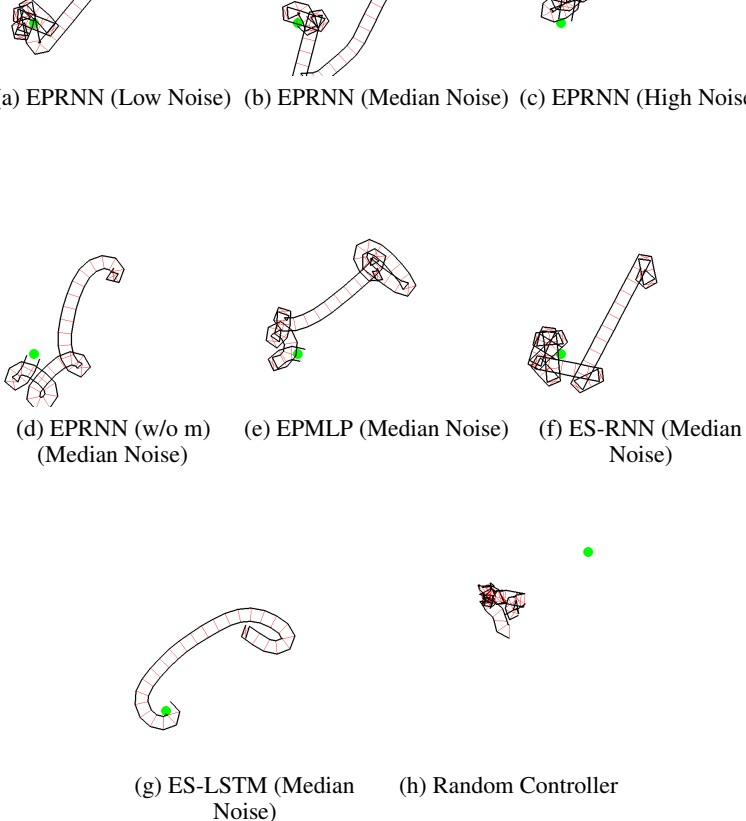

(a) EPRNN (Low Noise)  (b) EPRNN (Median Noise)  (c) EPRNN (High Noise)

(d) EPRNN (w/o m)     (e) EPMLP (Median Noise)   (f) ES-RNN (Median
(Median Noise)                                        Noise)

(g) ES-LSTM (Median       (h) Random Controller
Noise)

Figure 9: Trajectories of the agents from *Wheeled Robot Navigating* task with the task configuration of $(A = 0.5, k = 0.2, (g_x, g_y) = (-0.3, -0.3))$. We show the trajectories of the wheeled robot in black and red lines and the goal in green circles.

