# OpenReview forum: "Do What Nature Did To Us: Evolving Plastic Recurrent Neural Networks For Generalized Tasks"
_ICLR.cc/2022/Conference — ICLR 2022 Submitted_

### Official Review · Reviewer_byce · 2021-10-27

**Correctness:** 2
**Technical Novelty And Significance:** 2
**Empirical Novelty And Significance:** 1
**Recommendation:** 3
**Confidence:** 4

**Main Review:**

This paper proposes a new flavor of meta-learning, which is an interesting and promising research avenue currently trending. The proposed approach particularly aims at joining plasticity learning, evolutionary computation and recurrent neural networks (more trending topics), in a blend yet unpublished.
The state of the paper however is pretty rough, and it would require considerable work before it can be considered for publication in a major venue. The experimental setup particularly is inconclusive: the tasks are overly simplistic, custom but unmotivated, and addressed with an unnecessarily complex system, leaving to much room as to where positive or negative performances come from a specific component.

I would like to start by pointing out a few words or sentences taken from the paper that the authors should necessarily address -- just find them on the paper, I am confident the improvement can be evinced easily from the context. As a suggestion, focus on quantitative rather than qualitative remarks, and avoid claims unsustained in the literature (if not straight wrong).
- Researchers are increasingly obsessed...
- ANN have achieved great success [...] due to the strong capacity of handling large datasets
- Gradient oracles
- ANNs usually need overconsumption of datasets
- In a mimic manner
- Learning takes place in the feed-forward pass
- Recursion are found to be extremely sample-efficient
- The outer learning loop optimize the meta [hint: meta is an adjective, meta-learning is a type of learning, not "learning a meta"]
- [...] model to specific task by utilizing the meta [yep it was not a typo]
There are plenty more, but I am confident that the authors can take it from here. I strongly suggest to involve an English native speaker for general proofreading and edit, as this work could gain significantly from an exposition targeted to an international audience.
- It is found that evolution can be more efficient in cases of very long horizon in reinforcement learning [citing Salimans; this conclusion is not in the paper as far as I remember; would be more likely linked to the work by Stanley on open-endedness evolution]
- Meta learning aims at building learning machines that gain experience
- There are obviously bunches of more sophisticated neural structure
- Each result is concluded from independent 3 runs

Here is some feedback on the exposition. Also these are all requirements.
- It is unnecessary to italicize and underline the already Title Case words for the EPRNN acronym. Nor to bold it 7 times in the abstract. Nor in the rest of the paper.
- Some vocabulary which is uncommon for a ML audience and should be explained before usage, such as 'memristance' and biological 'plasticity'

Finally approaching the technical perspective: there are serious claims that need to be backed by a scientific process.
- Figure 1 illustrates a simplification of natural evolution as learning loop. The external loop is the hereditary passing of genetic material (left), the internal loop depicts the lifetime of an individual. The implication in this double-loop setup is that experience obtained by an individual through interacting with the environment is then passed genetically to its offspring, which incidentally would require self-modification of the DNA to incorporate the changes during the individual lifetime. Please confirm whether you stand by this interpretation, and provide citation to work done in verification and support to this hypothesis.
- The list of contributions shows 3 points. The first is a study that I could not find in he actual paper. The second describes the proposed meta-learning framework without clearly showing the contribution (which as far as I understand, is limited to having put these three pieces together, without even supporting the case as to why it should be advantageous). The third point is the fact that you run experiments (unless it is meant to highlight how the tasks were custom-made, though the paper should highlight the relevance and importance of these two tasks as it is unclear). In conclusion, the list of contribution needs re-writing to correctly represent a list of actual contributions sustained in the paper.

Reference to past work is incomplete (if not even biased) from a ML perspective. As a consequence the arguments on this paper are often unnecessarily complex or simply incorrect, again from the perspective of ML literature. This is also a requirement for publication.
- The literature review (and Table 1) should mention at the very least the work of Quoc Le and Esteban Real in AutoML, as it is the most active and recognized work in meta-learning of artificial neural networks.
- Evolutionary Algorithms are referenced from Zhang et al. 2011, pointing to a lesser known paper with a dubious title. The only evolutionary computation work that is utilized in multiple places is the OpenAI ES, a single paper published on ArXiv without peer review, which itself lacks a correct representation of the long-standing field of evolutionary computation. A few names to build a better awareness of the field include foundation work from Koza and Banzhaf, then Miikkulainen into Stanley, and absolutely Hansen or even Glasmachers.
- Even in something as fundamental as ANNs, the concepts of network architecture, recurrent connections, sample efficiency, gradient-based vs gradient-free, etc. are not as mysterious and vague as presented in this paper. I believe a more deep understanding of the underlying concepts can help the authors constructing a more compelling (and correct) argument.

As a concluding note: the authors should learn about neuroevolution of recurrent networks for continuous control tasks, as the literature has been going strong for over 20 years. Adding plasticity rules is not original per se, and wrapping into a meta-learning framework only automates hyperparameter optimization, itself minimal in modern evolutionary algorithms.
The proposed tasks are incredibly simple. As a requirement to change my mind on my recommendation, and to study the effective complexity of the tasks, I propose the authors to motivate extensively their need to construct their own custom tasks. As a baseline, I also see it necessary to include results from Random Weight Guessing of minimal-sized networks. For example, the first task (regression of a rescaled and translated sinusoidal with noise) can likely be solved by literally guessing (through subsequent random initializations) the weights of a neural network composed of only one layer of one recurrent neuron, or at worse one layer of 10 recurrent units followed by one feed-forward predictor. The architecture proposed in Figure 6 of two layers of 64 RELU sandwiching one layer of recurrent/plastic neurons is overly complex, highly inefficient, and not motivated.
The second task is even easier: a wheeled robot with power control on the wheels and a sensor estimating the distance from the goal. This class of problems is typically solved by a linear controller, as it requires no memory nor non-linearities, so I would expect to see the results of RWG on a network with just two feed-forward neurons (the ouput layer) again with no hidden layer. Once again the proposed structure, identical to the previous but for adding one extra 64-neurons fully-connected layer, is entirely out of scale for the problem, and unmotivated in the text.

**Summary Of The Paper:**

The authors put together their own flavor of meta-learning using off-the-shelf plasticity learning, evolutionary computation and recurrent network. Experiments are conducted on two overly-simplistic, custom-designed tasks, and the results lack impact.


**Summary Of The Review:**

While the introduction of yet another meta-learning framework should be supported, as the area is extremely promising, and the combination of plasticity learning rules plus evolutionary computation plus recurrent networks shows great potential, this paper is not in a shape proper for publication in a major conference, and the results on the overly-simplistic custom benchmarks are far from conclusive.

---

> ### Author Response · Authors · 2021-11-21
> **Reply to questions & concerns from Reviewer byce**
>
> 1. We agree that the tested environments are relatively simple, but we probably can not agree that a single neuron or linear controller could solve the two environments. We have performed investigations on the hidden layer size. With the naive recurrent neural networks, a network of sizes 16, 32, 64, and 128 yields apparent performance differences. We think it is important to emphasize that the robot can not directly observe the distance to the goal in the wheeled robot navigating task. It can only observe the signal strength, which is noisy and weakly related to the distance, dependent on some hidden configurations. Thus, it is not possible to solve this problem without memory, especially long term memory.
>
> 2. "Figure 1 illustrates a simplification of natural evolution as learning loop. …"
> There is certainly no self-modification of the DNA during the individual life span. The DNA (genetic information) corresponds to the META-PARAMETER, which will be passed to the next generation; The experience gained during each individual corresponds to the PARAMETER, which will be dropped instantly after its life, and will not be passed to its offspring. The reviewer may refer to self-modification as the "Fitness" arrow shown in Figure 1. Yes, the PARAMETER may affect whether the individual will have chances to have offspring and pass its DNA to the next generation. However, the PARAMETER will not change the META-PARAMETER, just like there will be no self-modification of the DNA.

---

> > ### Comment · Reviewer_byce · 2021-11-30
> > **Answer to comment**
> >
> > Thank you for your response.
> >
> > 1. I appreciate your disagreement, but running an experiment with a single neuron and random weight guessing (no need for training even) is so easy to run that I cannot see a reason not to definitely prove me wrong. I have however worked with neuroevolution for so long that I am keenly aware of the unexpected capabilities of such a simple approach. The trick here is that the larger the network, the larger the search space, so I have no surprise seeing that your experiments with a hidden layer have been unsuccessful, especially if you insist on recurrent units. Honestly, please, give it a try, not because I ask, but because understanding testbed complexity (and learning key techniques for such estimation) will undoubtedly improve your future work. It is frustrating when such a simple technique beats your new creation, I have been there, but this has always led me to significant quality improvements.
> >
> > 2. Glad we agree in that point, I now look forward to seeing the clarification in your updated submission.
> >
> > Just for clarification: I confirm that all of my other points still stand. I still support the original intuition underlying this work and I hope to see it fully explored in future work. Thank you.

---

### Official Review · Reviewer_RBhw · 2021-11-02

**Correctness:** 4
**Technical Novelty And Significance:** 1
**Empirical Novelty And Significance:** 1
**Recommendation:** 3
**Confidence:** 4

**Main Review:**

Strenghts:
- Optimizing learning rules is interesting and it's a promising approach.

Weaknesses:
- The proposed method is not new (the combination with recurrent neural networks may be, but it is still of very limited novelty).
- The language of the paper has many mistakes and is a bit informal in some parts ("researchers are increasingly obsessed", etc).
- The parallel Biological Neural Networks is interesting, but very weak.


**Summary Of The Paper:**

The authors explore the interesting (although not novel) topic of using derivative-free optimization / evolutionary algorithms to optimize learning rules.
The authors propose a method, EPRNN (Evolutionary Plastic Recurrent Neural Networks) that uses Evolution Strategies to learn a Hebbian learning rule for a recurrent neural network.

**Summary Of The Review:**

The main issues with the paper are language and the minimal novelty of the proposed methods.

---

### Official Review · Reviewer_oHzr · 2021-11-03

**Correctness:** 3
**Technical Novelty And Significance:** 2
**Empirical Novelty And Significance:** 2
**Recommendation:** 5
**Confidence:** 4

**Main Review:**

Strength:
The model appears to perform better than alternative models in the two tasks tested.

Weakness:
In my opinion, the main weakness of this work is the lack of intellectual insights or impressive empirical results. The main contribution is to apply evolutionary algorithm to meta-learn plasticity rules for RNNs. Given that there are quite a few papers in this area that involve two of the above elements (evolutionary algorithm, plasticity rules, RNNs), it is not too hard to combine the three.

Given the somehow arbitrary nature of the artificial tasks studied by the authors, it is difficult to tell whether the empirical results are impressive.

Minor concerns:
Intro: plastic rules --> plasticity rules
Intro: “plastic rules, aka Hebb’s rule”, Hebb’s rule is only one kind of plasticity rule
In general, the text can be edited to improve clarity
Should cite Confavreux Neurips 2020


**Summary Of The Paper:**

The authors applied evolutionary algorithms to meta-learn plasticity rules for recurrent neural networks. They show this approach performs better than alternative meta-learning approaches on two artificial tasks (sequence prediction and wheeled robot navigation).

**Summary Of The Review:**

The main challenge to this manuscript is the lack of intellectual insights, except a new combination of several existing techniques, and a lack of strong empirical results.

---

> ### Author Response · Authors · 2021-11-21
> **Reply to questions & concerns from Reviewer oHzr**
>
> 1. "Intro: plastic rules --> plasticity rules Intro: “plastic rules, aka Hebb’s rule”, Hebb’s rule is only one kind of plasticity rule In general, the text can be edited to improve clarity Should cite Confavreux Neurips 2020"
>
> Indeed, Hebb's rule is just one of the Plasticity Rule, we've updated in the paper

---

### Official Review · Reviewer_mN1j · 2021-11-03

**Correctness:** 3
**Technical Novelty And Significance:** 2
**Empirical Novelty And Significance:** 3
**Recommendation:** 3
**Confidence:** 4

**Main Review:**

The ideas behind the paper are interesting and investigating more biological-inspired methods that could potentially solve problems without gradient information are exciting. However, the contributions of this particular paper could be further elaborated on and the particular test domains are all rather simple. Beyond the approach introduced by Najarro&Risi (2020) and Soltoggio et al. (2018), the main addition is to also use a recurrent network. This by itself would be fine, but then the approach is only applied to a simpler RL domain than what Najarro&Risi investigated previously. Did the authors also try the approach on a more complex domain with higher input space? How did the approach perform there? That would be a more convincing demonstration of the power of RNNs+plasticity rules.

Additional comments:
- The visualization of what the weights learn are interesting. What did the recurrent hidden states learn?

- Why is the robot in Figure 3c navigating in these circles? It would be great to show trajectories before and after learning.

- Is the distance signal used as input to the neural network, or only used for the reward (as in the point nagiation task in the original MAML paper)? If used as a network input, how well would it do without it?

- "In some cases, it even produces better results compared with LSTM, despite the simpler model architecture. “ -> I’m not sure I would agree that the architecture is necessarily simpler.

- Why is only W_d mulitplied with the modulation signal in Equation 5?

- "Recursion-based learning employs recurrent neural networks (RNN), LSTM (Hochreiter & Schmidhuber, 1997), and self-attention (Mishra et al., 2018; Chen et al., 2021) layers as learners.” -> It would be great to have an exact definition of what the authors refer to as recursion.

- Figure 3 should mention what the different colors mean.

**Summary Of The Paper:**

In this work, the authors use evolutionary strategies to train recurrent neural networks with Hebbian plasticity rules. They test the system on two tasks, sequence prediction and a simple RL tasks that involve robot navigation. The approach is compared against previous work that uses plasticity but without recurrent connections and other approaches such as LSTMs. For the problems presented in this paper, the proposed approach outperforms most methods used in the comparison.

**Summary Of The Review:**

An interesting paper but currently the approach would need to be tested on more complex problems to more fully demonstrate the advantages of using an RNN+plasticity architecture.

---

> ### Author Response · Authors · 2021-11-21
> **Reply to questions & concerns from Reviewer mN1j**
>
> 1. "The visualization of what the weights learn are interesting..."
>
> We have added the visualization of hidden states in our updated version
>
> 2. "Why is the robot in Figure 3c navigating in these circles?..."
>
> Figure 3c is initially chosen for demonstration only. In our updated paper, we've shown those trajectories and added some comments.
>
> 3. "Is the distance signal used as input to the neural network, ..."
>
> The distance is only used as a reward (which is not used in observation.) Using the distance directly as observation would make this problem too easy to be solved by any algorithms. Instead of directly observing the distance, the cart only observes the noisy signals that are weakly correlated to the distance.
>
> 4. "Why is only Wd mulitplied with the modulation signal in Equation 5?..."
>
> Actually, every term is multiplied by modulation in Eq 5. Notice that the modulation term is absorbed into $h$, which gives $\hat{h}$ in Eq.5
>
> 5. "It would be great to have an exact definition of what the authors refer to as recursion."
>
> Indeed, it should be "Model-based learning" instead of "recursion-based learning."

---

> > ### Comment · Reviewer_mN1j · 2021-11-24
> > **Reponse**
> >
> > Thank you for the clarification. I still believe the work is interesting but needs some demonstrations on slightly more complex domains. Therefore I'm keeping my score for now but looking forward to seeing the approach being further developed and applied to other domains in the future.

---

### Decision · Program_Chairs · 2022-01-20

**Decision:**

Reject

**Comment:**

In this paper the authors demonstrate the use of meta-learning in plastic recurrent neural networks with an evolutionary approach, avoiding gradients. They show that this approach can be used to develop networks that can solve problems like sequence prediction and simple navigation.

The reviews for this paper all had scores below the acceptance threshold (3,5,3,3). The principal concerns were:

(1) The lack of novelty. Other papers have taken very similar approaches (e.g. Najarro & Risi, 2020 or Miconi et al., 2019), and fundamentally this paper simply ties together different elements in one package.

(2) Lack of demonstration of the approach beyond some very simple tasks.

(3) Lack of connection to the related literature on neuro-evolution and ML.

(4) General clarity and style of writing issues.

The authors responded to the reviewers, but the responses did not convince the reviewers enough to increase their scores past threshold. Given this, a reject decision was reached.